# Engineering Properties of Basalt Fiber-Reinforced Bottom Ash Cement Paste Composites

**DOI:** 10.3390/ma13081952

**Published:** 2020-04-21

**Authors:** Mohamad Hanafi, Ertug Aydin, Abdullah Ekinci

**Affiliations:** 1Department of Civil Engineering, European University of Lefke, North Cyprus, Mersin 10, Turkey; mhanafi@eul.edu.tr; 2Civil Engineering Program, Middle East Technical University, Northern Cyprus Campus, Kalkanli, Guzelyurt, North Cyprus, Mersin 10, Turkey; ekincia@metu.edu.tr

**Keywords:** bottom ash, basalt fiber, paste, strength, durability, sustainability

## Abstract

Extinction of natural resources builds up pressure on governments to invest in research to find more sustainable resources within the construction sector. Earlier studies on mortar and concrete show that bottom ash and basalt fiber are independently alternative binders in the concrete sector. This study aims to use bottom ash and basalt fiber blends as alternative novel-based composites in pure cement paste. The strength and durability properties of two different percentages of bottom ash (40% and 50%) and three volume fractions of basalt fiber (0.3%, 0.75%, and 1.5%) were used at three curing periods (7, 28, and 56 days). In order to measure the physical properties of the basalt-reinforced bottom ash cement paste composites flowability, dry unit weight, porosity, and water absorption measurements at 7, 28, and 56 days of curing were performed. Furthermore, the mechanical properties of composites were determined by unconfined compressive strength and flexural strength tests. Finally, to assess the durability, sulfate-resistance and seawater-resistance tests have been performed on composites at 28 and 56 days of curing. Results showed that the addition of basalt fiber improves the physical, mechanical, and chemical stability properties of paste up to a limiting basalt fiber addition (0.3% volume fraction) where, above, an adverse effect has been monitored. It is clear that observed results can lead to the development of sustainability strategies in the concrete industry by utilizing bottom ash and basalt fiber as an alternative binder.

## 1. Introduction

Rapid increments in urbanization and construction work have resulted in greater demand for construction materials, i.e., materials for which natural resources and energy are further consumed in order to manufacture, which results in harmful materials such as greenhouse gasses being produced in the process. These consequences harnessed the attention of scientists and governments toward the idea of sustainability or clean production, such as incorporating byproduct waste as a replacement for cement. Although intense research has been done and many developments have been made in the field of sustainable materials, they are still not widely used due to various reasons, such as the high processing costs and the lack of total understanding of the mechanical and engineering properties for these materials. However, according to statistics [1], civil works and building construction consume around 60% of raw materials extracted from the lithosphere and are estimated to use up to 40% of global energy consumption. Moreover, construction work has been found as one of the highest carbon emissions industries [2] due to the production, processing, and transportation of construction materials [3,4]. Previous studies estimated that 50% of CO_2_ emissions worldwide come from cement manufacturing sectors [5]. Recently, the tendency to replace such materials with sustainable admixtures such as waste is growing around the world. Some of the commonly used sustainable admixtures are bottom ash [6,7] and marble dust [8,9].

Coal bottom ash is a byproduct produced largely from coal-powered plants; thus, utilizing coal bottom ash in the concrete industry can be an economic and sustainable method for its disposal [10]. In general, bottom ash is believed to adversely affect the workability of concrete [11,12]. However, some published studies reported increased workability when using bottom ash as a replacement of natural sand [13,14]. Wongkeo et al. [15] investigated the effects of replacing Portland cement with BA showing that BA mixes had improved bulk density, thermal conductivity, and flexural and compressive strength. Furthermore, Aydin’s [16] study on the effects of adding BA to a pure cement matrix showed that composites with up to a 70% replacement of BA demonstrated suitable physical and mechanical properties to be used in construction.

Concrete is the main material used in the growing sector of construction. However, concrete is known for its brittleness, which inspired scholars to search for methods to alleviate this problem. One popular solution is the use of fiber reinforcement in a concrete matrix, which is a toughening material that has the potential to improve compressive strength as well as shearing and fracture resistance of concrete. Over the past years, developments have been achieved and a better understanding of the behavior of fiber-reinforced concrete has developed due to the vast research that was and is still ongoing in this field; more recently, these materials are being fabricated for hydraulic and civil buildings all around the world. Steel, carbon, glass, and polymer [17] are among the most popular types of these fibers. The most-used type of polymer fiber is polypropylene. The effects of these fibers on cement have been intensively studied. Many researchers [18,19,20,21,22,23,24] reported an increase in the compressive and flexural strength when adding fibers to cement composites. Valeria and Nardinocchi [18], for example, observed a reduction in the drying shrinkage upon adding fiber to cement composites. Moreover, Hwang et al. [25] reported enhanced flexural strength, toughness indices, plastic cracking, and impact resistance from the addition of natural fibers to cement composites.

Basalt fiber is one of the most popular fibers worldwide. It is a material that is usually made from the fine fibers of basalt. This fiber is similar in shape to glass fiber (GF). However, it has better physicochemical properties than GF; further, it has been reported to make better contributions to the properties of concrete [26]. In addition, the price of basalt is cheaper compared with carbon fiber, which makes it an ideal substitute for carbon fiber. More recently, basalt fibers have been widely used in civil and hydraulic engineering [26]. Regarding concrete, many studies have been conducted on the behavior of basalt fiber on the durability and strengths of concrete. Khan et al. [27] investigated the properties of concrete mixes enhanced with basalt and steel fibers, reporting an enhancement on the mechanical properties of concrete up to 0.68% of basalt inclusion. The authors also observed up to a 74% reduction in workability. Sun et al. [26] also investigated the addition of both short and long basalt fibers to concrete, finding that the compressive and splitting tensile strength of concrete increased with the addition of fiber up to 2% by volume and started to decrease after that, while bending strength kept increasing with increasing the fiber volume. The authors further found that short fibers were more effective in improving the strength of concrete. Sim et al. [28] reported that basalt fiber performed better than glass fiber under accelerated weathering conditions and provided higher resistance to temperature than that of glass fiber. Gamal et al. [29] conducted another study concerning the use of basalt fiber in concrete construction. They reported that the use of basalt fibers helped in retaining and improving the strength of concrete that is exposed to vegetable and mineral oils. The basalt-fiber-reinforced concrete could withstand the acidic, chemical, and salty effects that result in the reduction of the strength of concrete. Dong et al. [30] evaluated the potential of using basalt fibers to enhance the mechanical properties of concrete made with recycled earthquake waste. The authors also found that using basalt fibers can make up for the reduction in mechanical properties when increasing the ratio of waste replacement. Similarly, Wang et al. [31] suggested the use of basalt fiber with nanosilica to enhance the mechanical properties of recycled aggregate concrete.

Ahmad and Chen [32] studied the water and high-temperature resistance of mortars containing various proportions of basalt fibers and silica fume. They reported increased resistance by increasing the amount of silica fume and fibers as well as decreased porosity. Padalu et al. [33] investigated the use of basalt-reinforced mortar for wallettes strengthening. The strengthened wallettes showed four times higher strength, 29 times higher deformability, and 139 times higher energy-absorption capacity. Fenu et al. [34] also studied the dynamic behavior of mortars reinforced with basalt and glass fibers, investigating their influence on energy absorption and tensile strength. They reported increased energy absorption at high strains with the addition of fibers, while the dynamic increase factor was not significantly affected by the addition of fibers.

In this study, two different percentages of bottom ash (40% and 50%) and three volume fractions of basalt fiber (0.3%, 0.75%, and 1.5%) were used to investigate the physical, mechanical, and durability properties of the laboratory-produced composites. In the literature, to the best of our knowledge, no research consists of pure cement paste enriched with basalt fiber. This study was composed of comprehensive laboratory tests at three curing periods (7, 28, and 56 days). Additionally, the prepared composites were composed of bottom ash at high levels of utilization rate. The composites could be a promising alternative binder in the concrete sector and could be used as alternative novel-based composites. The durability properties of those composites were evaluated based on real-scale conditions. The samples were immersed in a seawater and sulfate solution to check their performance. In the literature, all studies focused on mortar and concrete properties; further, none of them were composed of pure cement paste enriched with basalt fiber. This research will fill the research gap in that particular area.

## 2. Materials and Methods

### 2.1. Materials

#### 2.1.1. Cement and Bottom Ash

Ordinary Portland cement CEM 1-42.5N, in accordance with the Turkish Standards Institution (TS EN 197-1), was used to prepare the cement pastes in this study. This cement has a Blaine fineness of 305 m^2^/kg and a specific gravity of 3.15. The loss of ignition for this cement is 2.5.

The used bottom ash was collected from a brick factory after the burning process. Coal was used for burning in the kiln at a temperature of around 1100 °C. The left-out ash was collected from the bottom of the kiln. Before being used, the bottom ash was sieved through the 1.18 mm sieve, and a fine powder was obtained. This powder was then dried in the oven at around 100 °C to ensure the absence of moisture. The resulting powder has an ignition loss of 3.9. The chemical compositions of these materials are provided in Table 1.

#### 2.1.2. Basalt Fiber

Basalt fiber used in this study was obtained from Dost Kimya Ltd., İstanbul, Turkey. The specific gravity of basalt fiber is 2.60 g/cm^3^. The length of the basalt fiber is 24 mm, and its diameter is 15 µm. Elastic modulus and tensile strength of the basalt fiber are 89 GPa and 4840 MPa, respectively. The elastic modulus measures the stiffness of the material and is related to atomic bonds and does not depend on strength. For quality purposes, generally tensile strength can be used. For this purpose, elastic modulus and tensile strength of basalt fiber were obtained from the manufacturer. Basalt fibers possessing higher tensile strength generally produce higher flexural strength. The experimental results also showed that basalt fibers have good compatibility with cement-based composites. Figure 1 shows the used basalt fiber.

### 2.2. Sample Preparation

A Hobart mixer (2.5 L capacity, WA, USA) was used to prepare pure cement paste composites. Bottom ash, cement, and basalt fiber were mixed in dry form for 30 s, and tap water was added slowly within 30 s. The fresh paste was placed into molds and then consolidated with a vibrating table within 1 min. After 24 h, the samples were removed from the molds and cured in water until testing ages (7, 28, and 56 days).

Castaldo et al. [35] used an uncertainty safety factor to overcome the effect of experimental uncertainties on estimations. This is important, especially for cement-based composites. Based on the safety concerns with buildings in the concrete construction sector without long-term performance data or information on novel-based composites produced here, there is reluctance to approve of those materials in the sector. The authors believe; however, that, for the sustainable building sector, using alternative industrial wastes to replace cement is mandatory for carbon dioxide reduction.

Additionally, as stated, the alternative approaches must be developed in building construction to minimize carbon footprints and give a chance to those novel binders that are readily developed in laboratory-scale research programs to make the world a better place. The presently available standards used in cement-based materials are not appropriate for those novel binders mainly produced from waste. Thus, those standards must be amended, or completely new test procedures can be adopted to the concrete sector.

Cubic molds, 50 mm^3^ in size, were used for the preparation of compressive strength samples. Mortar prisms, 40 mm × 40 mm × 160 mm in size, were used for the flexural strength samples. ASTM C109M-20 [36] standard for compressive strength and ASTM C348-19 [37] standard for flexural strength tests were used. Six samples for each curing age (7, 28, and 56 days) and for each mixture groups were produced for compressive strength and flexural strength tests. The digital compressive and flexural testing apparatus having a 200 kN capacity machine, which was designed for cement paste composites, was used. The testing apparatus was composed of rigid column frames with a hydraulic test chamber, LCD graphic digital control, and readout unit. Machines have recorded load-deformation with time. The central-point loading was used to measure the flexural strength. For both tests, failure load in kN is recorded and automatically converted to stress by dividing its area. The loading rate was 0.5 kN/s in compressive strength tests and the deformation rate was 0.05 mm/min in flexural strength tests.

The corresponding author has published articles [16,38] that cover 100% bottom ash and 100% pure cement. However, in this study, authors considered two main mixture groups as a reference: cement paste composed of 40% and 50% bottom ash as reference groups and enriched with three basalt fiber volume fractions (0.3%, 0.75%, and 1.5%). The testing program consists of two groups of mixtures, one with 40% bottom ash content (40-0 series), and the other with 50% (50-0 series). Each group includes various basalt fiber contents ranging from 0% to 1.5% by volume. The 40-0.3 corresponds to the mixture series composed of 40% bottom ash and 0.3% basalt fiber and the 50-0.75 corresponds to the mixture series composed of 50% bottom ash and 0.75% basalt fiber. The water-to-binder (w/b) ratio was kept constant for all mixture groups as 0.37 to attain the essential level of workability to prevent improper compaction.

Six samples were cast for each testing age (7, 28, and 56 days) and for each mixture group to measure the physical properties of the composites. The apparent specific gravity and water absorption experiments were performed according to the ASTM C127-15 [39] procedure. Consistency of the prepared mixtures was determined using a flow table test according to the ASTM C230M-14 [40] procedures. The sulfate resistance of composites was assessed by ASTM C88-18 [41] procedure and Ferraris et al. [42] study. Although the ASTM C88-18 [41] standard is known to measure the sulfate resistance of aggregates, the approach in this method was combined with the approach in Ferraris et al. [42] to measure sulfate resistance. The corresponding author has previously published publications on this modified method; further, we believe that this advanced approach is sufficient to measure the sulfate resistance of cement-based composites. The approach is to measure the weight losses of composites that continue until dispersed and to determine how much these losses differ from the main mass. The laboratory-produced composites were immersed in a 50 g/L Na_2_SO_4_ solution. The specimens were subjected to a sulfate solution until cracked. At the end of each cycle, the samples were removed from the sulfate solution and dried in an oven at around 105 °C. The mass changes were recorded in each cycle. The composites were immersed in seawater (obtained from the sea-shore directly) to simulate the real site condition and evaluate the performance of the laboratory-produced composites under real constraints. For seawater tests, the samples were immersed in seawater for one week, and the same procedure for sulfate tests was applied. The tests were continued until the first visible crack. The mass changes were then recorded. Six samples were prepared for each curing age (28 and 56 days) and for each mixture group to assess the composite resistance to sulfate and seawater. Figure 2 shows both flexural and cubic samples prepared at the laboratory that contain bottom ash and basalt fiber.

## 3. Results and Discussion

### 3.1. Effects of Basalt Fiber on Physical Properties

Figure 3 shows the flow values for basalt-reinforced bottom ash cement paste composites. Increasing the amount of bottom ash decreases the flow value beyond 0.3% basalt fiber. This is due to coarser particles of bottom ash. The interparticle friction hinders the movement. The addition of basalt fiber improves the flowability of the mixtures for both bottom ash mixture groups at a low volume fraction (0.3%). However, going beyond the 0.3% basalt fiber addition, the flow values in 40-0 mixture groups tend to decrease. The decrease in flow starts beyond 0.75% basalt fiber level in 50-0 mixtures. The absorption of available water, due to the large surface area of basalt fiber, caused a marginal decrease in flow values, especially beyond 0.3%. In addition, basalt fiber might absorb more cement paste at high volume fraction. However, the cohesiveness of the composites improved with the addition of basalt fiber. Similar results were found in Li et al. [43], Qin et al. [44], and Sadrmomtazi et al. [45]. Further, the composites are said to be workable if the flow values are approximately 150 mm [46]. In this study, all mixture groups except 40-1.5 mixture groups had flow values lower than 150 mm (containing 1.5% basalt fiber). However, this mixture group can also be easily cast into molds without extra workmanship. In this study, only one basalt fiber length (6 mm) was utilized. There are also 12- and 24-mm basalt fibers in length. One study performed on concrete showed that the increase in the length of basalt fiber results in a decrease in workability being more visible [47]. Based on the flow test results, all composites can be considered sustainable. It is believed that the addition of basalt fiber improves the consistency of the mix and decreases the energy requirements during casting. This could be a positive addition to basalt fibers, especially during formwork operations on-site; it also requires less workmanship. If the percentage of basalt fiber is kept at 0.3%, it will maximize the performance of the composites. Considering the 40-0.3 mixture groups, the increase in flow was calculated as 32% compared with the 40-0 mixture groups, and this increase was 11% compared with the 50-0 mixture groups. Additionally, the flow tends to decrease beyond 0.3% basalt fiber addition in 40-0 mixture groups, but this trend was beyond a 0.75% basalt fiber addition in 50-0 mixture groups. The 50-0 mixture groups show less reduction when compared with the 40-0 mixture groups. The reduction in flow at higher fiber volume fraction (1.5%) was approximately 32% for the 40-0 mixture groups and 14% for the 50-0 mixture groups. This might be due to the increments in fine material amount in 50-0 mixture groups, which contain less cement compared with that of the 40-0 mixture groups. When basalt fiber is used in large volume (1.5%), it produces greater shear resistance against the flow. Basalt fiber composed of filaments and large surface area requires more mixing water and more cement paste to coat all particles.

Figure 4 shows the dry unit weight (DUW) values for basalt-reinforced bottom ash cement paste composites at 7, 28, and 56 days of curing. The DUW values decrease with curing ages. The highest decrease (approx. 10%) was reported in the 40-1.5 mixture groups. The basalt fiber decreases the DUW values for both mixture groups when compared with the control sample. The average reduction for all series was about 7%. The decrease is more pronounced at high volume fractions. This might be due to the fact that, when basalt fiber is introduced into a system, it needs more paste to coat. This results in a reduction of composite density. The reduction in density may be linked to higher void content. In a homogeneous material, regardless of performed testing, the specific gravity is always constant. However, if a composite material is produced where the additive material has a lower specific gravity in comparison to the parent material, the matrix specific gravity reduces accordingly in relation to additive material content. It is a known fact that the pore structures within a matrix formed with hydration reaction. However, the decrease in DUW was higher in the higher cement amount group (40-0 series). This can also prove that more paste formation causes more reduction. The same findings can be found in [31,44]. The composites were classified as lightweight if the density is lower than 2000 kg/m^3^. Lightweight composites generally have good mechanical performance and superior durability. Based on the DUW values, all composites can be categorized as lightweight material; further, composites could be an alternative sustainable building material for the construction sector.

Figure 5 shows the porosity values for basalt-reinforced bottom ash cement paste composites at 7, 28, and 56 days of curing. Porosity decreases with curing ages. The rate of decrease increases beyond 28 days. This is due to the slow reaction of bottom ash particles. The addition of basalt fiber decreases the porosity at low volume fraction (0.3%). For beyond a 0.3% basalt fiber addition, the porosity values tend to increase. The compactness of the composites become less effective beyond 0.3% basalt fiber volume fraction, which can cause the formation of more voids. Additionally, basalt fiber increments require more cement paste, since the paste volume constant for all mixtures’ more porous matrix can form at high volume fractions. In addition, the decrease in the dispersion ability of fibers and cement paste can cause an increase in porosity values. The increase in water demand due to the addition of basalt fiber, especially at higher volume fraction, can be considered another factor for such an increase. At low volume fractions, the 40-0.3 and 50-0.3 mixture groups show 8% and 5% improvement at later ages, respectively. This improvement in porosity values was lower in the early curing period. This is expected due to the slower reaction of the bottom ash mixture groups. The increase in porosity was approximately 12% at early ages, which becomes 15% at later ages in 50-1.5 mixture groups. However, in 40-1.5 mixture groups, those values were 2% and 8% for early and later ages, respectively. The results are compatible with the water absorption (WA) test results. Compatible results can be found in [44,46,48,49].

Figure 6 shows the water absorption (WA) values for basalt-reinforced bottom ash cement paste composites at 7, 28, and 56 days of curing. The WA values tend to decrease with curing age. An increase in the bottom ash amount also increases the WA values. This can be explained by the fact that the addition of basalt fiber increases the pore connection at high volume fraction. Additionally, it can be seen from the flow values that the addition of basalt fiber decreases the flow values due to a higher absorption capability of basalt-enriched mixtures. The WA values tend to increase beyond the 0.75% basalt fiber addition level for both mixture groups. However, the increase in WA values is more in 50-0 mixtures compared with 40-0 mixtures. At early ages, the 1.5% basalt fiber addition caused a 16% and 18% decrease in WA values when considering the 40-0 and 50-0 mixture groups, respectively. This is due to the fact that bottom ash particles have a higher tendency to absorb more water. The 50-0 mixture groups were composed of a higher amount of bottom ash. At later ages, both mixture groups exhibit the same behavior. Similar findings were found in [31,45].

### 3.2. Effects of Basalt Fiber on Mechanical Properties

Figure 7 shows the unconfined compressive strength values for basalt-reinforced bottom ash cement paste composites at 7, 28, and 56 days of curing. The addition of basalt fiber increases compressive strength. However, this increase was diminished at beyond 0.75% basalt fiber addition. In addition, the increase in the capillary pores can cause a reduction in strength at a higher volume fraction of basalt fibers, as these fibers mostly affect the capillary porosity and not the gel porosity due to its large surface area; thus, at low volume fractions, more strength can be achieved. More gel formation via bottom ash particles in composites also helps in the increase in compressive strength. In this study, basalt fiber having 6 mm in length was used. This can also be a factor for strength increase. This length of basalt fiber seems to have better compactability and shows superior adhesion with cement paste. Similar findings are reported in [26,27,29]. The average increase from 7 to 56 days was approximately 27% for both mixture groups. The rate of increase was higher at low volume fraction. Beyond a 0.3% volume fraction of basalt fibers showed a lower rate of increase. This might be due to the weakening of the bond at high basalt fiber, increase of water absorption, and porosity of the composites. The rate of improvement in UCS was higher in the 50-0.75 mixture groups when compared with the 40-0.75 mixture groups for all curing ages. However, the 40-0.75 series showed higher strength at all ages. This is compatible with the porosity test results. When compared with the 50-0.75 mixtures, the increment in UCS was calculated as 65% at 56 days, and this was 57% for 40-0.75 mixtures at the same curing age. At a higher basalt volume fraction (1.5%), the UCS values significantly reduced by 51% at seven days for 40-1.5 series, and this reduction was 48% for the 50-1.5 series. When considering the UCS increment in 40-0 and 50-0 mixture groups at 7- and 56-day curing periods, this was 39% and 34%, respectively. When compared with the porosity values, basalt fiber decreases the porosity values by 10% and 9% in 40-0 and 50-0 mixture groups, respectively. Increasing the basalt fiber beyond 0.75% can cause a decrease in compressive strength. This may be due to the increase in the number of pores in the composites. The reduction in UCS values may be attributed to the decrease in flow values, which show nonuniform dispersion beyond 0.75%. At high volume fraction, the bonding force between basalt fiber and cement paste decreases. Loss of effectiveness to hold the matrix and development of the cracks at the fiber–matrix surface causes a reduction in UCS and FS values at high volume fraction (1.5%). The localized stress at the tip of the fiber hinders the pores, and more cracks can develop. Those cracks later opened and became larger. The bridging effect thus causes a sudden reduction in mechanical performance. As the additional level of basalt fiber increases, the amount of calcium ions in the system increases. This leads to the formation of more gel, which fills the available pores in the matrix. Thus, composites have more strength compared with that of without basalt fiber mixtures. Finally, one study [48] mentioned that the increase in compressive strength is due to the amorphous character of the basalt fiber.

Figure 8 shows the flexural strength (FS) values for basalt-reinforced bottom ash cement paste composites at 7, 28, and 56 days of curing. The same trend as observed in compressive strength was reported here. The addition of basalt fibers densified the matrix and better bonds were formed. The addition of basalt fiber seems to be effective until 0.75% volume fraction; beyond this value, a decrease in FS was reported. The loss of bridging effect and weakening of the matrix bonds between the cement pastes results in difficulty in compaction at higher rates and decrease in FS. Additionally, the addition of basalt fiber affects the fluidity of the composites and thus decreases the workability and FS at all curing ages. When considering the FS values, basalt fiber is more effective in 50-0.75 mixture groups compared with 40-0.75 mixtures at all curing ages. The improvement in FS was calculated as 65% and 57% for 50-0.75 and 40-0.75 mixture groups, respectively. The rate of increase in FS values was slightly lower (i.e., approximately 30%) for both mixture groups composed of no basalt fiber at the curing period between 7 and 56 days. At all volume fractions, both mixtures showed an increase compared with that of composites without fibers. The highest improvements were reported at 0.75% basalt mixture groups. At all additional levels, FS values tend to increase. The increase in FS values showed that a good bond exists between basalt fiber and cement paste. Another reason why there is no reduction in FS values beyond 0.75% may be due to the fact that the higher amount of basalt fibers is closely spaced with each other. Another effect may be due to the increasing amount of hydration products, which form better bonding with the basalt fiber. Additionally, at higher volume fraction, there are many fibers to bridge the action. Since basalt fiber is composed of silica and calcium oxide, it is expected to show better mechanical performance. The formation of calcium silicate hydrate gel leads to a higher compressive and flexural strength. Additionally, basalt fibers are in the form of an amorphous structure. This can help with the formation of a densified matrix with homogenous dispersion of basalt fiber in the cement paste. Saloni et al. [48] reported that basalt fiber acts as an aggregate and forms strong matrix properties, thus improving the strength. Similar results can be found in [44,45,46,48,50,51].

### 3.3. Effects of Basalt Fiber on Durability Properties

Figure 9 shows a sulfate-resistance test for basalt-reinforced bottom ash cement paste composites at 28 and 56 days of curing. Interestingly, sulfate-resistance improves with the addition of basalt fiber for all volume fractions. Actually, the resistance is governed by pore structure and amount of fiber. The addition of basalt fiber improves the bonding cement paste composites. The basalt fiber bridging mechanisms hold the matrix together and do not allow for weakening of the bonds. The rate of expansion decreases with increasing the amount of basalt fiber for both mixture groups. The average reduction of expansion due to the addition of basalt fiber was 30% for all mixture groups. The improvement might be due to the densification of the matrix and increments in FS values. This proves that sulfate ions did not result in adverse effects on basalt fiber. The increase in the bottom ash amount decreases the resistance to sulfate. Using 0.75% basalt fiber showed 44% less expansion at 28 days in the 40-0.75 series and 21% for the 50-0.75 series. When considering the same groups, if basalt fiber increased to 1.5%, those values were calculated as 51% and 46%. However, comparing the mechanical performance using basalt fiber beyond 0.75% reduces the compressive and flexural strength values. Similar conclusions can be found in [46,52].

Figure 10 shows the seawater-resistance test for basalt-reinforced bottom ash cement paste composites at 28 and 56 days of curing. The addition of basalt fiber decreases the weight loss by seawater and improves the chemical resistance of the composites. Negligible weight loss due to seawater was recorded for both mixtures and curing ages. The resistance becomes higher at later ages with the help of basalt fiber. The second control mixture group (50-0 series) has higher weight loss by seawater. However, the addition of basalt fiber diminishes this increase. The reason is mostly due to the addition of basalt fiber, which blocks the open-pore channels and connectivity of the pores and thus reduces the diffusion of ions into the composites. The 50-1.5 series is more resistant to sea attack compared with 40-1.5 mixture groups; the 40-1.5 mixture series showed 70% improvement at 56 days, and this value was increased to 88% for 50-1.5 mixtures series at the same curing period when compared with the without basalt fiber mixture groups. When considering the 40-0.75 mixture groups, the improvement was limited to 36% at 56 days. However, this was 79% for 50-0.75. There is little information available about the chemical stability of composites composed of basalt fiber [46,47,52].

## 4. Conclusions

Basalt fiber can be considered ecological material since it is manufactured from the basalt rock, which is naturally found in the Earth’s crust. Basalt is a natural rock found in abundance all around the world; it is used in the production of basalt fiber under a low-energy intensive process. Due to its high tensile strength and excellent modulus properties, its effectiveness in cement-based composites makes it an ideal candidate in the construction sector. Since it is considered an environmentally friendly material, the authors believe that its utilization will increase in the future. It will help to improve sustainability strategies for all nations. Here, in this study, the basalt-enriched bottom ash cement paste composites have proven their superior performance against sulfate and seawater attack. Additionally, composites have excellent mechanical properties. The use of bottom ash wastes to produce cement paste composites could possibly have a beneficial effect on the environment. The bottom ash used in this study can be considered as a high replacement level for cement (40% and 50%). The authors further believe that the composites can be sustainable and environmentally friendly. The experimental results showed that basalt fiber works well with bottom ash composites.

Although large portions of clays and other natural raw materials are used in masonry-related applications such as manufacturing of bricks, blocks, and paving units, the authors believe that use of natural raw materials consumes our resources; thus, bottom ash can be used effectively and satisfactorily in the building sector to reduce carbon dioxide emissions. Its utilization is still low, however, in the construction industry. Moreover, the use of bottom ash or other industrial waste in such applications, stated or controlled low-strength applications in the construction sector, has great potential in achieving a sustainable building industry and reducing carbon footprints. The authors further believe that using them in structural concrete applications provides a more sustainable approach in the concrete sector. However, more research should be conducted, especially for those applications to check safety regulations.

The authors checked the performance of their bottom ash cement paste composites through various tests such as physical, mechanical, and durability. Additionally, the authors evaluated the bottom ash cement paste composites according to the current international standards and tried to optimize the performance of our composites based on those mentioned tests. Based on the experiments in this study, the following conclusions can be reached:(1)The addition of basalt fiber improves the workability of the composites at a lower volume fraction. Beyond a 0.3% basalt fiber addition, the decrease in flow values was reported for all mixture groups.(2)The porosity of the composites increases as the basalt fiber volume fraction increases. The compactability of the fiber is adversely affected beyond 0.3% volume fraction.(3)The dry unit weight of the composites is classified as light weight. The produced composites have superior physical, mechanical, and chemical stability, which makes them an alternative sustainable construction material. Additionally, the mixture proportioning in this study can help for the development of sustainability strategies in the concrete industry by utilizing bottom ash and basalt fiber as an alternative binder.(4)The addition of basalt fiber increases the water absorption of both mixture groups beyond 0.3% volume fraction. More cement paste is needed when basalt fiber is introduced into the system. This affects the pore system of the composites.(5)The addition of basalt fiber increases the compressive and flexural strength. Both strengths tend to decrease beyond 0.75% volume fraction.(6)The addition of basalt fibers seems to be effective in chemical stability. Basalt fiber improves the resistance of the composites against sulfate and seawater.(7)Microscopic investigation should be conducted for a better understanding of these novel-based pure cement paste composites, which contain basalt fiber and industrial waste. The authors believe that the formation of calcium silicate hydrates and dispersion of basalt fiber in the matrix governs the overall behavior of the composites and need further investigation.

## Figures and Tables

**Figure 1 materials-13-01952-f001:**
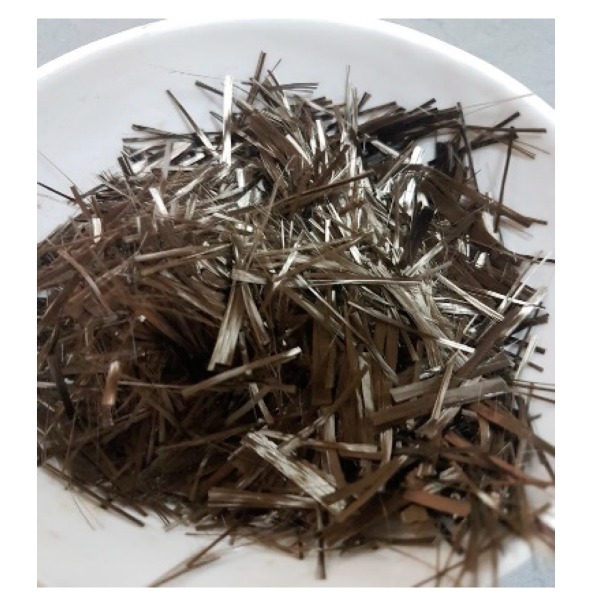
Basalt fiber used in this study.

**Figure 2 materials-13-01952-f002:**
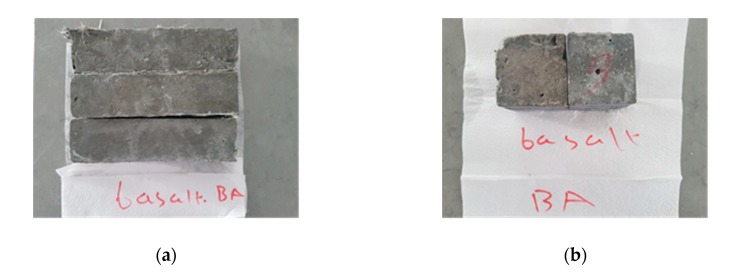
Basalt-fiber-enriched bottom ash composites: (**a**) flexural samples; (**b**) cubic samples.

**Figure 3 materials-13-01952-f003:**
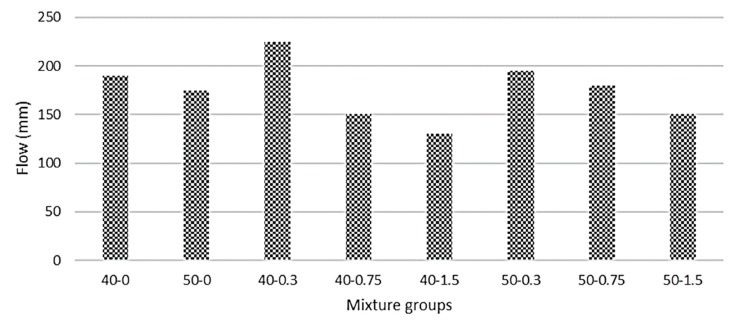
Flow values for basalt-reinforced bottom ash cement paste composites.

**Figure 4 materials-13-01952-f004:**
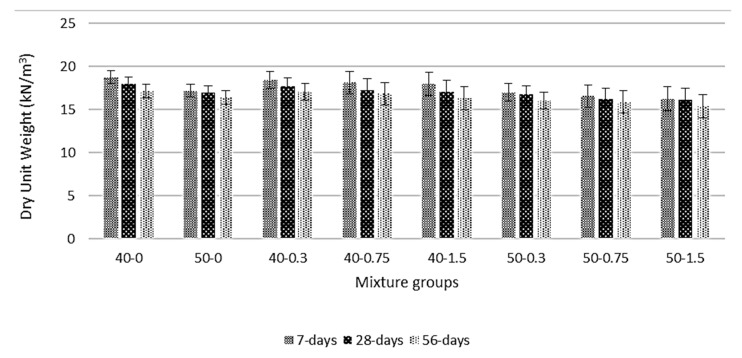
Dry unit weight for basalt-reinforced bottom ash cement paste composites at 7, 28 and 56 days of curing.

**Figure 5 materials-13-01952-f005:**
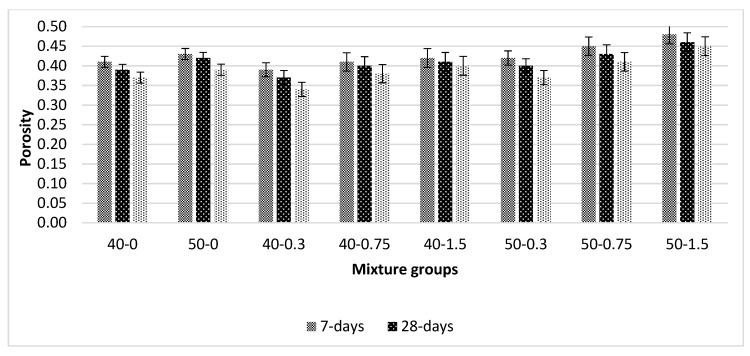
Porosity values for basalt-reinforced bottom ash cement paste composites at 7, 28 and 56 days of curing.

**Figure 6 materials-13-01952-f006:**
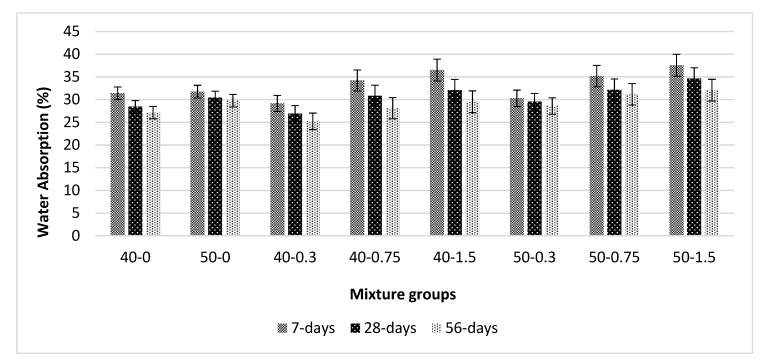
Water absorption values for basalt-reinforced bottom ash cement paste composites at 7, 28 and 56 days of curing.

**Figure 7 materials-13-01952-f007:**
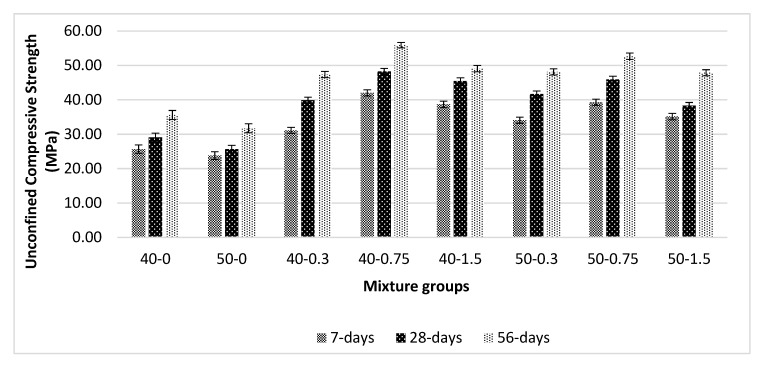
Unconfined compressive strength values for basalt-reinforced bottom ash cement paste composites at 7, 28 and 56 days of curing.

**Figure 8 materials-13-01952-f008:**
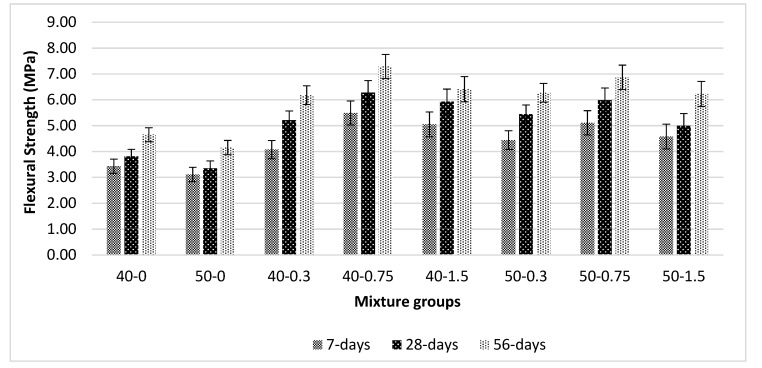
Flexural strength values for basalt-reinforced bottom ash cement paste composites at 7, 28 and 56 days of curing.

**Figure 9 materials-13-01952-f009:**
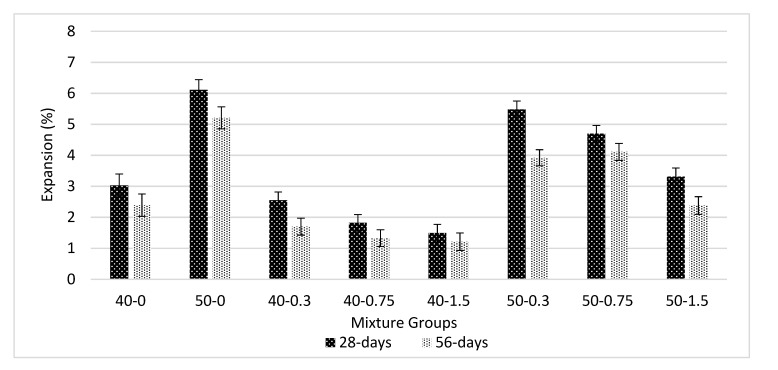
Sulfate resistance test for basalt-reinforced bottom ash cement paste composites at 28 and 56 days of curing.

**Figure 10 materials-13-01952-f010:**
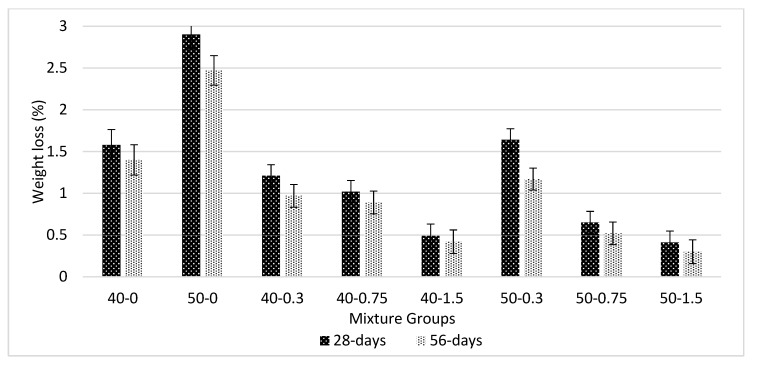
Seawater resistance test for basalt-reinforced bottom ash cement paste composites at 28 and 56 days of curing.

**Table 1 materials-13-01952-t001:** Chemical compositions of the both used cement and bottom ash.

Oxides (%)	Cement	Bottom Ash
SiO_2_	21.7	57.3
Al_2_O_3_	4.8	28.1
Fe_2_O_3_	3.9	6.1
CaO	63.6	1.4
MgO	0.3	0.2
K_2_O	0.3	0.8
SO_3_	1.4	0.7
LOI *^a^*	2.1	3.2

*^a^* Loss on ignition.

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
