# Peer review of "Engineering Properties of Basalt Fiber-Reinforced Bottom Ash Cement Paste Composites"

_materials, 2020, doi:10.3390/ma13081952_

Round 1

Reviewer 1 Report

Comments to the Authors

I have reviewed the manuscript “Fiber-reinforced cement paste composites for better sustainability” by Mohamad Hanafi, Ertug Aydin and Abdullah Ekinci. The work is well-organised and needs some revisions to be published in Materials. Here are my comments.

  1. Revise some typos in the text.

  1. Explain if the estimations are affected by experimental uncertainty as discussed in:

Castaldo, P., Gino, D., Mancini, G. (2019) Safety formats for non-linear finite element analysis of reinforced concrete structures: discussion, comparison and proposals, Engineering Structures, 193, pp. 136-153.

  1. Add some comments regarding the design possibilities to use the fiber-reinforced cement paste composites to improve the safety and sustainability of structures.

Author Response

Thank you for your suggestions. 

Reviewer 2 Report

The paper is focused on experimental investigation of physical, mechanical,
and durability properties of the composites composed of bottom ash and bazalt fibers.

In general, the paper is technically accurate. The objective of the study and the methodology followed are clearly outlined. The conclusions summarize the findings. The authors present a clear justification why the presented research is worth considering. I recommend to publish the paper in Materials.

Author Response

Thank you for your valuable suggestions

Reviewer 3 Report

Manuscript ID: materials-759045  

Title: Fiber-reinforced cement paste composites for better sustainability

Any action aimed at reducing the consumption of natural resources, as well as reducing the amount of waste is beneficial to the environment and human health. Therefore, the research topic can be considered as up to date and is in line the recent trends in construction materials technology.

General comments to work:

Introduction:

Introduction part is written well, the reviewed literature is well organized and described. Scope of work is well-founded and presented correctly. This is a strong part of this manuscript.

Materials and methods:

The article requires quality improvements in the part presenting the research methodology.

verse 120 – CEM I, add compressive strength class

verse 134 – 89 GPa, on what basis were elastic modulus and tensile strength adopted?

verse 165 – figure 3 – signed fig. 1 (verse 186)

verse 187 – figure 4 – signed fig. 2 (verse 199)

verse 201 – figure 5 – signed fig. 3 (verse 213)

verse 215 – figure 6 – signed fig. 4 (verse 223)

verse 226 – figure 7 – signed fig. 5 (verse 242)

verse 244 – figure 8 – signed fig. 6 (verse 254)

verse 257 – figure 9 – signed fig. 7 (verse 268)

verse 280 – figure 10 – signed fig. 8 (verse 280)

verse 157 - The Authors did not provide information about the concentration of sodium sulfate and the seawater. ASTM C88-18 refers to aggregate testing. There is no adequate justification why this method was used.

verse 196 – what is the basis for saying that the composites can be classified as light materials?

Author Response

thank you for your valuable comments and your contribution

Reviewer 4 Report

In my opinion, the manuscript does not present sufficient quality to be published. Therefore, my assessment is "rejected".

Below are a few comments that have led me to my decision:

- The title does not correspond to the content. I’m afraid that the word 'sustainability' in the title is there solely because it is a topical and modern concept. The described experiment and its outcomes have little in common with sustainability.

- A fundamental error of the experiment is the absence of reference material - cement paste without bottom ash. The effect of partial replacement of cement by bottom ash (and possibly its influence on sustainability, which has already been mentioned in the title by the authors) could not be determined in any way.

- All figures are extremely unclear. This also relates to unnecessarily long designations of materials (designations for example 40-0, 40-0.3, 40-0.75, etc. would suffice).

- There is no data on results variability - standard deviation, coefficient of variation, etc. Statistical analysis of results is also completely missing. Without these data, the results provide very little relevant information.

- The manuscript does not explain the choice of the water binder ratio of 0.37.

- More detailed information on the testing of the monitored parameters (e.g. strength parameters) is missing – what standards or procedures for testing of the pastes were employed, what instruments, etc.

- Information on flow values (lines 175 and 176) does not correspond to the results (Fig. 3).

- The manuscript insufficiently explains why the pastes were tested according to standards ASTM C127-15 and ASTM C88-18, which are designed for aggregate testing. Density is typically determined in the case of pastes - dry unit weight, on the other hand, is given for soil. The procedure for determining the sulfate resistance can be modified depending on the tested material.

- It is unusual that both the specific gravity (dry unit weight) and porosity decrease with the duration of ageing in the case of all materials.

- The manuscript contains a large number of formal errors.

Author Response

Thank you for your suggestion and valuable contribution to us. 

Round 2

Reviewer 1 Report

Comments to the Authors

I have reviewed the revised manuscript “Fiber-reinforced cement paste composites for better sustainability” by Mohamad Hanafi, Ertug Aydin and Abdullah Ekinci. The work is well-organised and can be published in Materials.

Author Response

Thank you very much for your valuable contribution in our paper.

Reviewer 3 Report

Note to Authors

L. 134 - add unit to the value of 89 (elastic modulus)

Author Response

thank you for your valuable suggestion.

Reviewer 4 Report

At first glance, it is evident that the quality of the revised manuscript is significantly higher than of the original version. Nevertheless, I do believe it is not entirely without errors and I have several comments on the revised version:

  1. I still consider it a pity that the authors did not include a cement paste without bottom ash as a reference material. Obviously, it should contain the same amount of basalt fibres as the bottom ash pastes (0, 0.3, 0.75 and 1.5%). It is a pity especially since they probably have the data available. Nevertheless, I understand and accept the situation.
  2. It is commendable that the authors included variability of results. However, I am confused by the provided information. The text "The coefficient of variation of the mixture without fiber is 1.38%. The coefficients of variation for 0.3%, 0.75%, and 1.5% of the basalt fiber mixture groups are 1.79%, 2.34%, and 2.41%, respectively. The standard deviation for compressive strength and flexural strength is calculated as 0.37 and 0.04 MPa, respectively, for mixed groups without fiber. The standard deviation for basalt fiber groups is 0.88 MPa, 0.90 MPa, and 0.93 MPa for 0.3%, 0.75%, and 1.5%." indicates that the authors merged materials 40-X and 50-X into one statistical set. That would constitute a fundamental mistake. The standard deviation (eventually the coefficient of variation) must be calculated for each material separately - i.e. the value for 40-0, another value for 50-0, another for 40-0.3, etc. I also recommend unifying the variability indicators for the individual tests - either use the standard deviation or the coefficient of variation (or both) for all the results. The above-cited text does not provide a clear idea for which property the coefficients of variation have been calculated - for sulphate resistance or for resistance to seawater? It is also unclear for which strength of the fibre pastes the standard deviations have been calculated - I assume for compressive strength. If so, what are the standard deviations for the flexural strength?

Therefore, I recommend adding unambiguous values of the indicators of result variability to all properties and to the individual test sets – e.g. by inserting standard deviations into graphs in the form of error bars. Alternatively, to use box plots instead of bar charts.

3. The manuscript still contains minor formal errors. For example, it seems that the text includes different line spacing styles (lines 144-161 have larger space between them than most of the remaining text, while lines 120-128 have smaller space between them). The manuscript also contains different formatting of figure captions - the caption of Fig. 1 is aligned centrally, while the captions of the other figures are not. Nevertheless, I expect these formal errors to be eliminated within the final stage of manuscript publication.

Author Response

special thanks to you. without your suggestion and contribution we are not able to reach this final stage. when we look at our first draft and this, as you mentioned in your review comments "significant improvement" thank you again
